# BMI at Discharge from Treatment Predicts Relapse in Anorexia Nervosa: A Systematic Scoping Review

**DOI:** 10.3390/jpm12050836

**Published:** 2022-05-20

**Authors:** Stein Frostad, Natalia Rozakou-Soumalia, Ştefana Dârvariu, Bahareh Foruzesh, Helia Azkia, Malina Ploug Larsen, Ehsan Rowshandel, Jan Magnus Sjögren

**Affiliations:** 1Division of Psychiatry, Haukeland University Hospital, 5021 Bergen, Norway; 2Research Unit Eating Disorders, Psychiatric Center Ballerup, Maglevænget 32, 2750 Ballerup, Denmark; natalia.rozakou.soumalia@regionh.dk (N.R.-S.); stefana.darvariu@regionh.dk (Ş.D.); baharehforuzesh@gmail.com (B.F.); helia.azkia.01@regionh.dk (H.A.); malina.ploug.larsen@regionh.dk (M.P.L.); ehsanrowshandel@gmail.com (E.R.); magnus.sjogren@umu.se (J.M.S.); 3Department of Clinical Science, University of Umeå, 90185 Umeå, Sweden

**Keywords:** anorexia nervosa, predictor, relapse, weight gain, systematic review, eating disorders

## Abstract

Background: Anorexia nervosa (AN) has high rates of enduring disease and mortality. Currently, there is insufficient knowledge on the predictors of relapse after weight normalization and this is why a systematic literature review was performed. Methods: PubMed, EMBASE, PsychInfo, and Cochrane databases were searched for literature published until 13 July 2021. All study designs were eligible for inclusion if they focused on predictors of relapse after weight normalization in AN. Individual study definitions of relapse were used, and in general, this was either a drop in BMI and/or reccurrence of AN symptoms. Results: The database search identified 11,507 publications, leaving 9511 publications after the removal of duplicates and after a review of abstracts and titles; 191 were selected for full-text review. Nineteen publications met the criteria and included 1398 AN patients and 39 healthy controls (HC) from adults and adolescents (ages range 11–73 years). The majority used a prospective observational study design (12 studies), a few used a retrospective observational design (6 studies), and only one was a non-randomized control trial (NRCT). Sample sizes ranged from 16 to 191 participants. BMI or measures of body fat and leptin levels at discharge were the strongest predictors of relapse with an approximate relapse rate of 50% at 12 months. Other predictors included signs of eating disorder psychopathology at discharge. Conclusions: BMI at the end of treatment is a predictor of relapse in AN, which is why treatment should target a BMI well above 20. Together with the time to relapse, these outcomes are important to include in the evaluation of current and novel treatments in AN and for benchmarking.

## 1. Introduction

Anorexia nervosa (AN) is a disabling, costly, and potentially deadly illness that typically commences during adolescence and is clinically characterized by extreme weight loss or failure to gain weight appropriately for age. Prevailing symptoms usually are an intense fear of weight gain, often with a fear of fattening food, relentless control of food intake, and distorted body image [1]. The outcome of AN is poor, although, in the past 20 years, an increasing number of studies have been performed, whereof some are gaining empirical support for efficacy [2,3,4]. During the acute stage, weight normalization is the foundation of treatment, while in the long term and in order to achieve behavioral change, outpatient psychotherapy is usually provided [5]. In children and adolescents, family-based psychotherapy is validated/endorsed by evidence [6]. At the same time, there is inconsistent scientific support for the use of cognitive behavior therapy and other psychotherapies in adolescents and adults [4,5,7,8]. A major issue is the high rate of relapse after completing treatment [5] and the substantial proportion of patients that develops severe and enduring AN (SE-AN) [9]. Both for research and clinical praxis, another issue is that the definition of relapse in AN varies substantially [10]. Studies have reported a wide range of estimates of the rate of relapse after treatment in AN, depending upon the definitions of remission and relapse used, the length of follow-up, and the interventions employed. For example, normal weight restoration typically precedes the normalization of pathological eating disorder symptoms by a year [11]. In addition, differentiation of relapse due to incomplete weight normalization treatment with exacerbation and relapse after full weight normalization may be difficult. In a systematic review on relapse by Khalsa et al., the authors suggested a definition of relapse as BMI ≤18.5 or ≤85% ideal body weight with significant fear of gaining weight or disturbance in body image combined with restricting, bingeing, or purging behaviors for one month. Should the symptoms persist for 3 months, this would constitute a full relapse [10]. Commonly reported relapse rates after outpatient psychotherapy ranged from 9% to 70% [12,13,14], with relapse rates after hospital admissions ranging from 50% to 70% in some studies [14,15]. Furthermore, time to relapse is seldom described and is of relevance as an outcome when studying the treatment effects of various interventions. One prospective study described the time to relapse after weight normalization as being 17 months on average [16], while another prospective study found it to be 4–9 months [12]. There is a substantial risk that incomplete weight normalization and consequent exacerbations lead to a cycle of recurring treatment failures and recurring relapses that, with time, contribute to the development of severe and enduring AN. Thereby, it is essential to elucidate predictors of relapse after complete weight normalization to enable an increased awareness of and possibly, to adapt and intensify treatment efforts when these predictors are present. Since there is a lack of collated knowledge of the predictors of relapse after complete weight normalization, we consequently conducted a systematic review of the scientific literature to investigate predictors of relapse in AN after weight normalization, with targeted treatment of normalization defined as BMI ≥ 18.5 kg/m^2^. We present the results in a narrative systematic ‘scoping’ review format, where identified predictors of relapse are presented.

The purpose of the present systematic scoping review was to investigate predictors of relapse as well as time to relapse after weight normalization as defined by bodyweight with BMI ≥ 18.5 kg/m^2^. The characteristics and quality of the identified studies are described as a systematic, narrative review.

## 2. Materials and Methods

The protocol was written and registered with the International Prospective Register of Systematic Reviews (PROSPERO): Id number CRD42021265393 (“Factors influencing time to relapse after weight restoration treatment in Anorexia Nervosa”). We employed the Preferred Reporting Items for Systematic Reviews and Meta-Analyses guidelines (PRISMA) [17].

### 2.1. Eligibility Criteria

Studies were independently examined for eligibility by several authors (E.R, H.A, B.F, M.P.L, N.R.-S., S.D., J.M.S. and S.F.) according to the PICOs criteria (population, intervention, comparison, and study design) [17]. The population (P) of interest was adolescents or adults with AN, and outcomes (O) were (a) predictors of relapse and (b) rate of relapse in AN. We included studies published before July 2021 in English or Danish. The following types of studies were eligible: original publications of case series, randomized controlled trials, cross-sectional studies, and cohort studies. Intervention (I) was weight normalization, and investigations into predictors of relapse after medication or after psychotherapeutic intervention were not considered within the scope of this review. In adults, completed treatment with BMI ≥ 18.5 kg/m^2^ or BMI-percentage corresponding to 18.5 kg/m^2^ or above were regarded as criteria for remission [1,18]. In children and adolescents, remission was regarded as weight corresponding to a BMI percentile above 14 percent or a minimum of 95% of the expected ideal body weight (IBW) for sex, age, and height as determined by the Centers for Disease Control and Prevention Growth charts [19,20]. Any comparator (C) was eligible. Inclusion criteria were: all types of AN, all ages and genders, human studies, post-weight restoration treatment, and all types of study designs, studies with predictors of relapse after weight normalization. Exclusion criteria were: animal studies, reviews, editorials, commentaries, and non-AN.

This systematic review made use of each individual study’s own definition of relapse into AN, as long as it was made according to established diagnostic criteria, e.g., according to DSM-V [1].

### 2.2. Search Strategy

The literature search was conducted on 13 July 2021 and covered material/literature available in the electronic databases PubMed, EMBASE, PsycINFO, and Cochrane. The key terms were modified according to the database and included: “Anorexia nervosa”, “Weight restoration”, “Prognosis”, and “Time to relapse”. In addition, the reference lists of all included publications were screened to identify additional eligible publications. Further information on the search terms is available in the Appendix A.

### 2.3. Study Selection

The selection of relevant publications is summarized in the PRISMA flow diagram (Figure 1). After searching the databases, references were imported into EndNote X20. Following the removal of duplicates, the remaining publications were imported from EndNote into Rayyan QCRI [19]. Using Rayyan QCRI’s ‘blind mode’, titles and abstracts were screened independently by at least two authors, and the final set of publications was thereafter double-checked by two authors (N.R.-S. and S.D.). Reviewers rated whether studies met inclusion criteria by (a) reviewing titles and abstracts and thereafter (b) conducting a full-text review. The selection of individual papers was conducted in accordance with the predetermined inclusion and exclusion criteria. In cases of a dataset being used in multiple or overlapping publications, we selected the one with the newest date or with the largest dataset. The size of the dataset was prioritized over the newer publication date in case of conflict. Any disagreement was resolved through consensus discussion with an experienced psychiatrist and scientist (J.M.S.).

### 2.4. Data Extraction and Synthesis

To address the two research questions of this review (identification of predictors of relapse and time to relapse after weight normalization), available data (i.e., author name, publication year, study design [cross-sectional, intervention, randomized control trial], number of included individuals with AN or healthy control (HC) sample, age and gender, weight restoration treatment, intervention and follow-up duration, potential predictors of relapse, biological predictors, psychological predictors, ED-related predictors, and time to relapse) were extracted from the included papers (see the full table in the Appendix A). All data were cross-checked by two reviewers (N.R.S. and S.D). Since some of the included publications omitted raw data on time to relapse, attempts were made to contact the authors to try to retrieve these missing data. If data were not provided or authors did not respond, studies were excluded from the systematic review. Missing values on descriptive data did not lead to the exclusion of studies.

### 2.5. Risk of Bias in Individual Studies

To evaluate the quality of the included publications, a bias assessment was conducted. Since the publications differed in terms of study design, two different tools were used: The Risk of Bias in non-randomized Studies tool (ROBINS-I) [21] was used to evaluate the risk of bias in non-randomized control trials (NRCTs). The assessment of each type of bias was categorized into four groups (low, moderate, serious, and critical) based on seven main domains: bias due to confounding, bias in the selection of study participants, bias in classifications of interventions, bias due to deviations from intended interventions, missing data bias, measurement of outcome bias, and bias of selected results. For observational studies, the Quality in Prognosis Study tool (QUIPS) [22] was used. Six domains were assessed for bias with this tool, including study participation, study attrition, prognostic factor measurement, outcome measurement, study confounding, and statistical analysis and reporting. Each domain could be considered as low, moderate, or high bias. For every study, regardless of the bias assessment tool used, the overall risk of bias was determined based on the ratings of each bias domain and was rated as low, medium, or high.

## 3. Results

### 3.1. Study Search and Selection

The database search identified 11,507 publications, leaving 9511 publications after the removal of duplicates, as illustrated in Figure 1. Following the screening of abstracts and titles, 9320 publications were excluded. The remaining 191 studies were reviewed for eligibility in full text, according to the inclusion and exclusion criteria. A total of 19 studies were ultimately included in the current narrative review.

### 3.2. Characteristics and Quality of the Included Studies

The 19 studies included 1398 AN patients and 39 HC from both the adult and adolescent populations, with the participants’ ages ranging from 11–73 years. Sixteen out of nineteen studies exclusively comprised female patients, while the remaining three included a small percentage of males (<33.3%). Most studies were conducted in the USA (9 studies), and the rest were in Canada, Germany, France, and Italy. Two of the studies were conducted in a combination of sites in the USA and Canada. All studies focused on weight restoration, and the interventions included multidisciplinary treatment programs (nutrition rehabilitation, weight restoration, and psychotherapy), inpatient treatment, intensive group therapy, or behavioral programs. The potential predictors of relapse in the included publications were characterized as either biological, psychological, or ED-related. More details on the characteristics of the 19 studies can be found in Table 1 and Appendix A.

#### 3.2.1. Study Design

Most of the included studies used a prospective observational study design (12 studies; 1056 patients), a few followed a retrospective observational design (6 studies; 342 patients), and only one of the studies was a non-randomized control trial (NRCT) [23]. The patients in the prospective and retrospective studies had similar age (mean ±/− SD were 24.6 ± 4.9 versus 24.4 ± 2.2 years, respectively), duration of AN (6.1 ± 3.5 versus 6.8 ± 2.08 years), and severity as expressed by BMI at the start of treatment (15.0 ± 0.80 versus 15.5 ± 1.1 years, respectively). The follow-up duration ranged from 2 months to 26 months, with the exception of one study which failed to mention the time point of participant reassessment [24].

#### 3.2.2. Sample Size and Considerations for Statistical Analyses

Most studies included a small to a medium number of participants and ranged from 16 to 191 participants. Generally, power calculations were not conducted in the statistical analyses of the included studies, except for one study which investigated the predictive power of the selected prognostic factors [25]. Population size was briefly considered in the interpretation of the results, as most studies reported its potential impact.

#### 3.2.3. Bias Assessment

Regarding the quality of evidence, only two studies (~10.5%) were of low quality, while nine studies (~47%) were of medium quality, and eight studies (42%) were of high quality (see Figure 2 and Appendix A). The attrition domain was at the highest risk of bias, with five studies (26%) at moderate risk, and five (26%) at high risk, adding up to over 50% of the included studies. The confounding domain also represented an issue, as eight studies (42%) were at moderate risk of bias, and four (21%) were at high risk. Most studies were at moderate risk of confounding due to the ambiguity in the measurement, as well as the plan to address potential interfering factors. In contrast, statistical analysis and measuring the outcomes were domains where studies had a low risk of bias, 73% and 78%, respectively. Lastly, the measurement of prognostic factors and selection of study participants presented an overall low risk of bias, 68% and 52%, respectively. The one study that was assessed with ROBINS-I presented an overall low risk of bias, with only the confounding and selection of participants’ domains as the sources of moderate risk of bias [23]. A comprehensive summary of data from eligible studies is presented in the data extraction table (Appendix A).

#### 3.2.4. Review Question 1: What Is the Time to Relapse?

Twelve of the nineteen studies presented relapse rates for different time periods, ranging from 2 months [26] up to 26 months [16]. The studies had different designs and purposes. One study presented relapse rates of 27% at 2 months [26], some reported 21–31% at 6 months [27,28], while other studies found relapse rates in the range of 30–72% at 12 months [16,24,29,30,31,32,33,34,35,36,37]. Half of the studies reported a relapse rate of approximately 50% at 12 months (M ± SD: 47.4 ± 19.2%). The studies reported duration of illness, age, the severity of illness, and comorbidities as potential factors influencing the time to relapse.

#### 3.2.5. Review Question 2: Which Are the Predictors of Relapse?

The predictors were classified into three groups: biological predictors, psychological predictors, and ED-related predictors. Of the 19 studies included in this review, 8 excluded patients with a BMI below 18.5 kg/m^2^ at discharge [12,16,30,32,33,37,38,39]. Another eight studies had not excluded patients with BMI below 18.5 kg/m^2^, but the variability of BMI at discharge indicated that potentially a minor subgroup of these patients was not in complete remission [23,25,26,31,34,35,36,40]. In addition, three of the studies may have included a significant minority of unremitted patients, but the design of the study could still allow for relevant data on outcome prediction [24,27,28].

##### Biological Predictors

In total, 13 studies concerned biological predictors of relapse, and 8 of these described BMI or weight as a predictor of relapse [24,28,30,31,32,34,39]. Only one of these studies had a high [31], most others moderate, and only one had a low quality of evidence [39]. The studies differed in design, BMI measurement time points, and whether weight or/and BMI were assessed. However, all studies supported that low BMI, either close to or within the normal range, at the end of treatment was disadvantageous, leading to a higher risk of relapse.

Three studies investigated the percentage of body fat as a predictor of relapse [30,33,38], and while two found a low percentage of body fat to be a predictor of relapse [33,38], one did not [30]. Two studies had a high quality [33,38] and one a low quality of evidence [30].

Three studies investigated levels of leptin at discharge, and two high-quality studies found it to be a predictor of relapse [26,33], while one moderate quality study did not [25].

Two studies aimed at identifying radiological predictors of relapse. One moderate quality study on adults found that reduced improvement of MRI changes in the right dorsal anterior cingulate cortex from admission to discharge predicted relapse [35]. A high-quality fMRI study on adolescents concluded that hypoactivation of the medial prefrontal cortex at discharge was associated with poor outcomes at one-year follow-up [23].

##### Psychological Predictors

Three studies found psychological predictors of relapse of relevance; one moderate quality study found a history of suicide attempt, previous ED treatments, the intensity of obsessive symptoms, and residual body shape concerns to be predictors [16]. A high-quality retrospective study found low self-esteem to be predictive of relapse [36].

##### ED-Related Predictors

Moderate quality studies found exercise immediately after discharge [16], as well as BP-AN subtype, the severity of body-checking behavior (pre-treatment), decrease in motivation to recover (during treatment), lower motivation to recover (post-treatment) [12], and the EDE score to be predictors of relapse [40]. Studies of high quality showed that normative eating self-efficacy [27] and diet energy density [37] were predictors of relapse.

##### Study Design

Comparing study designs, there was no statistical difference in relapse rate at 12 months between prospective and retrospective studies (F = 1.7 n.s.).

#### 3.2.6. Summary of Evidence

Based on the high quality of evidence and low risk of bias, the 19 included studies indicate that the strongest predictor of relapse in AN after weight normalization is a lower BMI (within the normal range). Evidence of lower quality suggests that decreased amounts of body fat and leptin levels, as well as ED and psychiatric symptoms such as suicidality and low self-esteem at discharge, can be predictors of relapse in AN after weight normalization.

## 4. Discussion

This systematic review identified 19 publications relevant for the aims of describing the time to relapse after weight normalization treatment in AN and predictors of this relapse. Of these 19 publications, 12 found a relapse rate between 30–72% over a time period of 2–26 months. Half of the studies (6 out of 12) reported a relapse rate of approximately 50% at 12 months, and calculating the mean of all these studies confirmed this. One study that followed the patients for 26 months found that the relapse rate leveled off at 17 months [16]. The study by Berends et al. [41] found the highest risk of relapse during the first year and thereafter, a maintained risk for up to two years. Time to relapse may be affected by several factors such as duration of illness, age, severity, the proportion of AN binge-purge subtype, presence of comorbidities, and potentially other factors. However, the number of eligible studies and the diversity of outcomes assessed precluded a more thorough analysis of factors that could potentially modify the time to relapse. Thereby, the main finding on time to relapse is that 50% of patients with AN relapse into the disorder 12 months after achieving weight restoration.

With regard to predictors of relapse in AN after weight-restorative treatment, BMI at the end of treatment emerges as the main predictor. Although the studies differed in many ways, this systematic review finds that not reaching a sufficiently high BMI, within the normal range, at the end of treatment will increase the risk of relapse. This finding is informative as it will help to set a target for the patient and therapists in their common ambition to reach remission. In addition, for those underway, it will help inform at what stage of recovery the patient may be, i.e., when treatment takes longer time, and it will thereby help to set expectations. Interestingly, although only supported by a narrative comparison, among patients who achieved a BMI above 20 at the end of treatment, somewhat contradictory findings were observed with, e.g., one study indicating that BMI does not affect relapse risk [37], while another finding that BMI still is a predictor of relapse in this high BMI range [32].

Two other findings from this review, albeit of low level of evidence, that low body fat percentage was a predictor of relapse [30,33,38] and that low leptin levels at discharge predicted risk of relapse [25,26,33], both harmonize with the finding that low BMI is a predictor of relapse, together with supporting that achieving a sufficiently high BMI also implies that the body fat percentage has increased and thereby leptin levels, at the end of treatment is a treatment goal that creates the most optimal protection against relapse.

Two studies included in this systematic review, using either MRI or fMRI of the brain, identified possible radiological signs associated with relapse risk. Reduced improvement of MRI changes in the right dorsal anterior cingulate cortex from admission to discharge predicted relapse in one study [35]. Persisting hypoactivation of the medial prefrontal cortex at discharge was associated with poor outcomes at one-year follow-up in adolescents [23]. The neuronal circuits of these brain regions are part of a network involved in decision making [42] and visual working memory [43], both of which may be relevant for recovering from AN. A failure to return to normal functioning in these brain regions may influence recovery both in the short term and if sustained inducing plasticity induced changes in the wiring of the brain [44]. Clearly, including brain imaging in research on AN to increase our knowledge about treatment effects and predictors of relapse seems a worthwhile ambition to consider. Thereby, future studies may inform whether radiological assessment can be used during the last stages of treatment to determine whether treatment should continue to help reduce the risk of relapse.

The number of studies presenting psychological and ED-related predictors of relapse after weight normalization was scarce, preventing any strong conclusions from being made. However, findings such as the intensity of obsessive symptoms, body shape concerns, excessive exercise immediately after discharge [16], low motivation to recover [12], normative eating self-efficacy [27], and Eating Disorder Examination (EDE) score [40], together stresses the necessity of a complete remission, i.e., including ED symptoms for improving the prognosis. Obviously, the disease is not confined to an anthropomorphic outcome but is by definition a psychiatric disorder with a range of psychological mechanisms that need to be addressed in order for the disorder to be sufficiently and successfully treated. Any treatment that only improves BMI while leaving other symptoms and signs of the disorder unaddressed will be associated with a substantial risk of relapse. The current status of knowledge, at least in adults, has not identified any such treatment that efficiently and with lasting effects restores normal both BMI and psychological functioning in AN. Thereby, a continued effort to characterize changes in AN over time, including effects of treatment and risk factors of relapse, is essential and will create a framework for directing the design of future studies. One important task for the AN research community is to agree on common standard outcomes to be used in all studies enabling benchmarking.

Other findings of psychological factors associated with the risk of relapse include a history of suicide attempt [16] and low self-esteem [36]. These results may be relevant in the context that affective symptoms influence the prognosis of AN, with depression being both the most common comorbidity and the one factor that most clearly influences the prognosis of AN [45]. Attempting to address affective symptoms in AN, whether related to a comorbid affective disorder or reflective of a struggle to cope with the negative effects of the disease, should be prioritized in the treatment of AN. Only one study found that the binge-purging subtype of AN had a higher relapse risk [12] which was a surprise since it frequently indicates a more active disorder. In addition, it may also suggest that the disorder is more dynamic and that there is an influence of bulimic psychopathology in this subset of patients. However, in the lack of studies further exploring this potential risk factor, any far-reaching conclusions cannot be made regarding neither binges nor the intensity of the disease.

Most studies (12/19) in this review were prospective observational studies, and only six studies were retrospectively designed. The populations were similar, and both the prospective and the retrospective studies found that BMI at the end of treatment was a predictor of relapse into AN [24,28,30,31,35,40]. Thereby, taking study designs into account, BMI still emerged as a predictor of relapse. Obviously, a prospective study is preferred in view of the risk of bias in a retrospective study; however, the fact that both types of study design found similar results further stresses BMI as a predictor of relapse.

In view of the nature of the disorder, with the inherent ambivalence to weight gain felt by patients with AN, how would the results of this study benefit patients with AN and their carers? Part of the therapeutic approach is to provide information about the disease, backed by science, to patients and carers to enable them to consider options and support in decision making. Apart from the ambivalent nature of the disease, or perhaps as an integrated feature of this, patients frequently have erroneous beliefs [46] about what they may benefit from, including their diet, body, and physical activity. Thereby, providing information to help in correcting these beliefs is one step in support of recovery. In addition, it provides clinicians with scientific support in the ambition to reach sufficiently high BMI in their patients, to prevent relapse into AN.

There are a few limitations of this study, one being that some studies may have included a minor group of unremitted patients [24,27,28]. It is thereby possible that a minor influence on the risk of relapse came from patients that were not completely remitted at the end of treatment. The ambition of this systematic review was not to compare different definitions of remission but rather to investigate predictors of relapse after weight normalization. The results clearly demonstrate the need to treat until remission as defined by a normal BMI, and preferably above BMI 20 kg/m^2^, since not reaching this anthropomorphic state is associated with a risk of relapse. Another potential limitation was that data did not allow for statistical analyses, for example performing a meta-analysis on BMI values or other potential predictors of relapse. This would have helped in increasing the strength of evidence for this narrative review. The ambition of the current study was to do a meta-analysis, which failed due to the lack of raw data and variations in follow-up times, outcomes, and definitions of relapse. Future studies should aim to share data and harmonize study designs to allow for a meta-analysis on time-to relapse and predictors of relapse to be conducted. Yet, another limitation is that the influence of different types of treatments was not assessed in relation to the risk of relapse. This is an important field of research for the future. Furthermore, several potential psychological predictors of relapse were described in different studies, but most of these stemmed from studies including a limited number of patients suffering from AN, and very few studies were able to reproduce these results. Since predictors related to social functioning [23], self-esteem [36], motivation to recover at discharge [12], diet energy density, and diet variation [37] are all potentially modifiable risk factors for relapse, it is encouraged to further investigate these factors in upcoming studies.

We had the ambition to do a meta-analysis of the results; however, two groups were needed for this type of analysis, and there was insufficient data on mean and standard deviation on remitters or non-relapsed to allow for this analysis, e.g., on time to relapse. For the same reason, a meta-regression could thereby also not be carried out. We also considered a Pearson correlation, but in view of the lack of raw data, this was not feasible. A regression analysis was also considered but rejected due to the same reasons as for not performing a Pearson correlation. Finally, correlation coefficients from the individual studies were missing. Some statistics were included in the results above for crude estimations, e.g., on time to relapse.

## 5. Conclusions

This systematic scoping review found evidence that a lower BMI (within the normal range) at the end of treatment is a predictor of relapse in AN after weight normalization. This was supported by other findings such as low levels of body fat and low leptin levels after weight normalization treatment, which also were predictors of relapse as well as being associated with BMI. In addition, the presence of ED symptoms at the end of treatment indicated a higher risk of relapse, thereby underscoring the relevance of complete remission for prognosis. Including the rate of relapse and time to relapse in future randomized clinical trials will help provide data for benchmarking and enable comparisons of current and novel interventions.

## Figures and Tables

**Figure 1 jpm-12-00836-f001:**
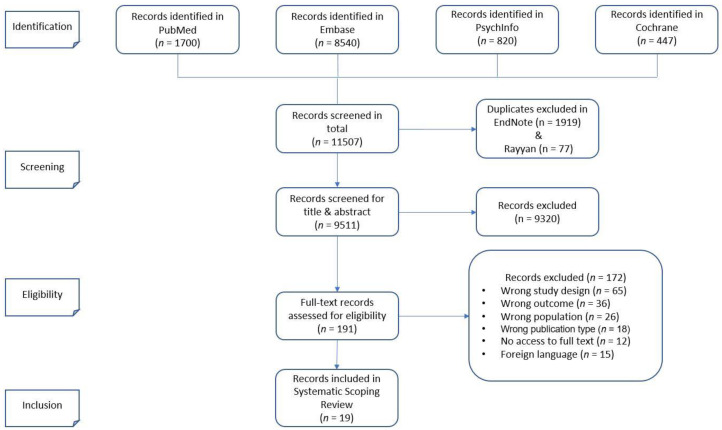
Prisma flow diagram of study screening and inclusion.

**Figure 2 jpm-12-00836-f002:**
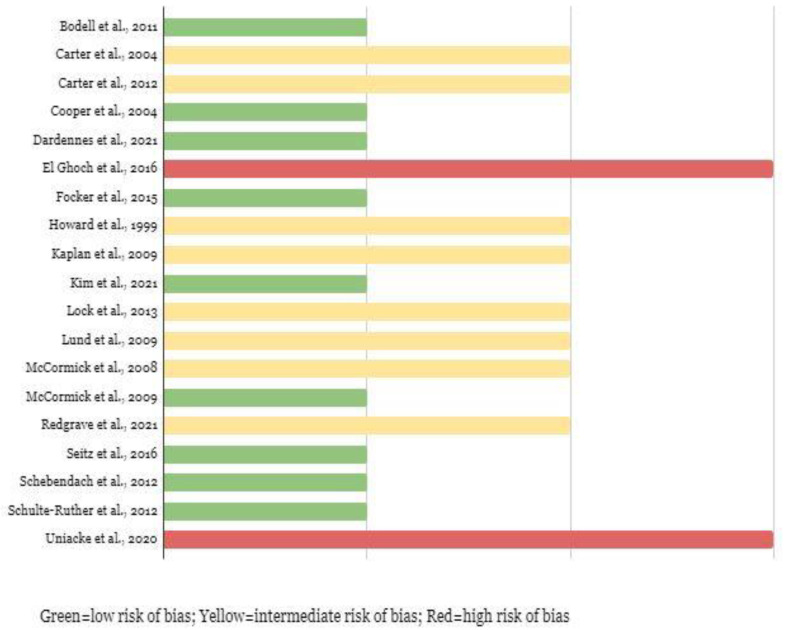
Risk of bias assessment for every individual study.

**Table 1 jpm-12-00836-t001:** Characteristics of included studies.

First Author, Year	Study Design	Sample Size and Diagnosis	Age, Duration, Severity *, Gender and Country	Intervention and Follow-Up Duration	Definition of Relapse, and Results	Key Findings (Predictors of Relapse Bolded)
Bodell and Mayer, 2011	Observational-Prospective	*n* = 21 (AN)	M ± SD = 26.6 (5.5) D: NA S: NA F = 100%; M = 0% USA	Structured behavioral program (weight and eating behavior normalization) FU: 6–9 mo	Def: MROS TTR: NA Relapse rate: 52%	- **Percent body fat** was significantly lower in the poor outcome group.
Carter et al., 2004	Observational-Prospective	*n* = 51 (AN)	M ± SD = 26.9 (9.0) D: 6.2 ± 6.8 years S: BMI = 15.1 ± 2.1 F = 100%; M = 0% Canada	Intensive group therapy program (weight and eating behavior normalization) FU: 15 mo	Def: BMI < 17.5 for 3 mo. TTR: M ± SD = 17.0 (4.1) mo Relapse rate: 35%	- **History of suicide attempt(s)** and previous specialized treatment for ED, the **severity of obsessive compulsive symptoms** at presentation, **excessive exercise** immediately after discharge, and **residual concern about** **shape, weight, and eating** at discharge were all significant predictors of relapse.
Carter et al., 2012	Observational-Prospective	*n* = 100 (AN)	M ± SD = 25.4 (7.7) D: 6.3 ± 7.2 years S: BMI = 15.1 ± 1.9 F = 95%; M = 5% Canada	Multidisciplinary: (1) Nutritional rehabilitation (2) Weight restoration (3) Eradication of binge (4) Eating and purging (5) Group psychotherapy (6) (CBT, DBT, and IPT) FU: 12 mo	Def: BMI < 17.5 for 3 mo. TTR: 4–9 mo Relapse rate = 41%	- **Low motivation to recover** at discharge, as well as change in motivation from admission to 4 weeks, BP subtype of AN, and **high body-checking behavior** at admission, were all predictors of relapse. - **High EDE-Q Eating** at discharge and **high EDE-Q Eating, Shape and Weight concern**, along with a **history of childhood physical** **abuse**, also predicted relapse to a lower extent.
Cooper et al., 2021	Observational-Prospective	*n* = 146 (AN)	M ± SD = 30.1 (14.39) D: 10.2 ± 11.2 years S: BMI = 15.6 ± 1.8 F = 92.5%; M = 7.5% USA	Behavioral meal-based protocol to help patients interrupt unhealthy eating and weight control behaviors FU: 6 mo	Def: BMI < 19 at FU. TTR: 6 mo Relapse rate: 23% non-responders: 18%	- **Weight restoration** at discharge, **normative eating self-efficacy** at admission, and less steep improvement in this from admission to FU were associated with weight restoration at FU.
Dardennes et al., 2021	Observational-Prospective	*n* = 26 (AN)	M ± SD = 26.5 (4.3) D: Fair outcome group: 7.6 ± 5.5 years, poor outcome group: 9.1 ± 3.7 years S: BMI = 14.5 ± 1.6 F = 100%; M = 0% France	Behavioral nutritional rehabilitation and weight restoration program FU: 2 mo	Def: BMI < 18 at 2 months FU. TTR: NA Relapse rate: 27%	- **High leptin levels** at discharge were significantly correlated with high BMI at FU.
El Ghoch et al., 2016	Observational-Prospective	*n* = 54 (AN)	M ± SD = 25.3 (7.4) D: 7.9 ± 6.3 years S: BMI: 15.6 ± 1.7 and 14.8 ± 1.6 in treatment success (TS) versus treatment failure (TF) group F = 100%; M = 0% Italy	Inpatient CBT-E and day care before discharge FU: 12 mo	Def: MROS TTR: 12 mo Relapse rate: 52%	- No significant difference in total body fat or trunk fat percentage between relapsed and remitted patients. - **High BMI at discharge** significantly predicted 1-year normal weight maintenance.
Focker et al., 2015	Observational- Prospective	*n* = 161 (AN)	M ± SD = 15.2 (1.5) D: 1.0 ± 0.7 years S: BMI = 15.1 ± 1.3 F = 100%; M = 0% Germany	Day patient treatment after short inpatient care or inpatient Treatment FU: 12 mo	Def: Readmission to inpatient treatment. TTR: 12 mo Relapse rate: 20%	**BMI-percentile at discharge** significantly predicts BMI percentile at 1-year follow up.
Howard et al., 1999	Observational-retrospective	*n* = 59 (AN)	M ± SD = 24.8 (8.7) D: 5.0 ± 6.1 years S: 16.0 ± 2.0 F = 100%; M = 0% USA	Transferred from inpatient treatment to a day hospital program FU: NA	Def: Readmission to inpatient treatment. TTR: NA Relapse rate: 24%	At the time of day, hospital admission **BMI below 19** significantly predicted day hospital treatment failure.
Kaplan et al., 2009	Observational-Prospective (from RCT)	*n* = 93 (AN)	M ± SD = 23.3 (4.6) D: 4.5 ± 3.6 years S: NA F = 100%; M = 0% Canada and USA	Behavioral weight restoration program FU: 12 mo	Def: BMI < 18.5 TTR: 6 months Relapse rate: 57% TTR: 12 mo Relapse rate: 72%	- Greater **normalization of right dorsal anterior cingulate cortex grey matter prospectively** predicted sustained remission.
Kim et al., 2020	Observational-retrospective	*n* = 41 (AN)	M ± SD = 25 (5.3) D: 7.4 ± 6.1 years S: NA F = 100%; M = 0% USA	Inpatient treatment (weight restoration and reduction of psychological distress) Involved medical management, psychotherapy, and dietary intervention. FU: 12 mo	Def: BMI < 18.5 TTR: 12 mo Relapse rate: 51%	- **Percent body fat, fat-adjusted leptin, and high log leptin** independently predicted weight maintenance at 1 year.
Lock et al., 2013	Observational- retrospective (from 5 RCT)	*n* = 111 AN 83 adolescents 28 adults	M ± SD = 20.2 ± 4.0 D: NA S: BMI adolescents = 16.1 ± 1.1, BMI adults = 18 ± 2.1 F = 100%; M = 0% USA and Canada	(1) Adolescent AN: FBT AFT (2) Adult AN: CBT, (3) medication and a (4) combination FU: 12 mo	Def: BMI ≤ 19 TTR: NA Relapse rate: NA	- Achieving a **bodyweight of 95.2% of expected body weight by EOT** is the best predictor of recovery for - adolescents with anorexia nervosa. - For adults with AN, the most efficient predictor of weight recovery was **weight gain to greater than 85.8% of ideal body weight**. The most efficient predictor of psychological recovery was the **achievement of an EDE weight concerns scores below 1.8**.
Lund et al., 2009	Observational-prospective	*n* = 79 (AN)	M ± SD = 21.6 ± 7.7 D: 4.8 ± 6.3 years S: BMI = 16.3 ± 1.7 F = 100%; M = 0% USA	Inpatient treatment (weight restoration and reduction of psychological distress) Involved medical management, psychotherapy, and dietary intervention. FU: 12 mo	Def: Increase in CGI-S during the 1-year FU. TTR: 12 mo Relapse rate: 41%	- **Weight gain rate during inpatient treatment** for AN was a significant predictor of short-term clinical outcomes after discharge.
McCormick et al., 2008	Observational- retrospective (from NRCT)	*n* = 18 (AN) *n* = 10 data on follow up	M ± SD = 25.6 (7.24) D: 6.5 ± 5.3 years S: BMI = 13.5 ± 2.1 F = 66.7%; M = 33.3% USA	Inpatient treatment FU: 12 mo	Def: BMI < 18 TTR: 12 mo Relapse rate: 70%	- The best predictors of weight maintenance in weight-restored AN patients over 6 and 12 months were the level of **weight restoration at the conclusion of acute treatment** and the **avoidance of weight loss immediately following intensive treatment**.
McCormick et al., 2009	Observational-retrospective	*n* = 20 (AN)	M ± SD = 27.6 (9.45) D: 10.5 ± 8.5 years S: BMI = 16 ± 2.3 F = 100%; M = 0% USA	Inpatient treatment FU: 12 mo	Def: Readmission to a partial or inpatient unit and/or BMI < 17.5 TTR: 12 mo Relapse rate: 35% Unknown: 25%	- **Improved self-esteem (MMPI-2)** predicted better outcomes, while **worsening low self-esteem** predicted relapse.
Redgrave et al., 2021	Observational- Prospective	*n* = 191 (AN or OSFED) -follow up data for n= 99 *n* = 166 (AN) *n* = 25 (OSFED)	M ± SD = 32.55 (12.29) D: 13.2 ± 11.6 years S: BMI = 16.2 ± 2.1 F = 100%; M = 0% USA	Behavioral weight restoration program FU: 6 mo	Def: Sustain BMI < 19 for 6 months FU. TTR: NA Relapse rate long-term ill: 21% Relapse rate short-term ill: 31%	- **BMI at discharge** was the only significant predictor of maintaining at least a BMI of 19 kg/m^2^ at follow-up. - Duration of illness was not associated with a BMI ≥ 19 kg/m^2^ at follow-up.
Schebendach et al., 2012	Observational-Prospective	*n* = 19 (AN) (16 in the analysis)	M ± SD = 25.8 ± 3.8 D: 6.3 ± 2.9 and 4.0 ± 2.5 years for treatment success (TS) versus treatment failure (TF) group, respectively S: 15.5 ± 1.4 versus 14.6 ± 1.4 for TS versus TF groups F = 100%; M = 0% USA	Multidisciplinary: (1) Structured behavioral program (normalizing weight, and eating behavior) (2) Individual (supportive, cognitive, and behavioral elements), group and family therapy (3) Weight restoration FU: 12 mo	Def: MROS TTR: 9–12 mo Relapse rate: 50%	- **Lower DEDS prior to hospital discharge** is associated with worse outcomes after inpatient treatment. - Non-caloric fluid and fat intake predicted DEDS. - Carbohydrate, protein, and non-caloric fluid did not predict the energy density score.
Schulte-Ruther et al, 2012	NRCT	*n* = 19 (AN) *n* = 21 (HC)	M ± SD = 15.7 ± 1.5 D: 1.0 ± 0.7 years S: BMI = 15.3 ± 1.5 F = 100%; M = 0% Germany	Multimodal treatment program: (1) Nutritional rehabilitation (2) Weight management (3) Cognitive-behavioral therapy on an individual and group basis, and family-based interventions. FU: 12 mo	Def: MROS TTR: NA Relapse rate: NA	- **Hypoactivation in the brain network** supporting the theory of mind may be associated with a **social–cognitive endophenotype reflecting impairments of social functioning** in anorexia nervosa, which is predictive of a poor outcome at 1-year follow-up.
Seitz et al., 2016	Observational-Prospective (from RCT)	*n* =121 (AN)	M ± SD = 15.6 (1.5) D: 0.9 ± 0.7 years S: BMI: 15 ± 1.3 F = 100%; M = 0% Germany	Stepped care program of stabilizing inpatient treatment Randomized to inpatient or day-patient care FU: 12 mo	Def: Readmission to hospital following inpatient treatment discharge. TTR: NA Relapse rate: 17%	- Serum leptin levels and weight gain rate did not predict age-adjusted BMI at follow-up.
Uniacke et al., 2020	Observational- Retrospective (from RCT)	*n* = 93 (AN)	M ± SD = 23.3 (4.6) D: 4.7 ± 3.7 years S: BMI = 15.4 ± 1.8 F = 100%; M = 0% USA and Canada	Behavioral weight restoration program FU: 12 mo	Def: Weight maintenance: the BMI never fell below 18.5 kg/m^2^ for four consecutive weeks TTR: 6 mo: Relapse rate: 57% TTR: 12 mo: Relapse rate: 72%	- Neither weight suppression nor its interaction with BMI predicted successful weight maintenance at 6-or 12 months, or time to relapse.

* Severity described as BMI at start of treatment. Abbreviations: AFT = Adolescent Based Therapy, AN = Anorexia Nervosa, BMI= Body Mass Index, BP = Binge/Purge, CBT = Cognitive Behavioral Therapy, CBT-E = Enhanced Cognitive Behavioral Therapy, CGI-S = Clinical Global Impression-Severity, D = duration, Def = Definition of relapse into AN, DBT = Dialectical Behavioral Therapy, DEDS = Diet Energy Density Score, Def = Definition, EOT = End of Treatment, ED = Eating Disorder(s), EDE = Eating Disorder Examination, EDE-Q = Eating Disorder Examination-Questionnaire, F = Females, FBT= Family Based Therapy, FU = Follow-Up, IPT = Interpersonal Therapy, HC= Healthy Controls, M = Males, M ± SD = Mean ± Standard Deviation, MMPI-2 = Minnesota Multiphasic Personality Inventory-2, mo = months, MROS = Morgan-Russel Outcome Scale, NA = Not Available, NRCT = Non-Randomized Control Trial, OSFED = Other Specified Feeding and Eating Disorder, RCT = Randomized Control Trial, S = severity at end of treatment, TTR = Time To Relapse.

## Data Availability

Data in this publication is available in a publicly accessible repository that does not issue DOIs at: https://docs.google.com/spreadsheets/d/1iIqXynVELW4_GeiBhzLWYeeKiV6OC0UQxoUJN-rqDCw/edit?usp=sharing (accessed on 20 February 2022) and at https://docs.google.com/spreadsheets/d/1iIqXynVELW4_GeiBhzLWYeeKiV6OC0UQxoUJN-rqDCw/edit?usp=sharing (accessed on 20 February 2022).

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
