# Peer review of "BMI at Discharge from Treatment Predicts Relapse in Anorexia Nervosa: A Systematic Scoping Review"

_jpm, 2022, doi:10.3390/jpm12050836_

Round 1
Reviewer 1 Report
This paper describes a systematic review focusing on predictors of Anorexia Nervosa relapse. This is an interesting topic, giving an important overview of the literature. However, I do belief some aspects could be further clarified to give a reader a better interpretation of the results:
- As the authors focus on relapse, a clear definition of relapse should be given, including clear in- and exclusion criteria related to this definition.
- Additionally, each included study used their own definition of relapse. These definitions needs to be described to help interpreting the results and drawing further conclusions.
- In the description of study interventions, insights on duration, intensity, etc. should be given.
- The first time the research questions “what is the time to relapse” is mentioned is in section 3.2.4. this RQ should be introduced earlier already. Again, the answers given here depend on the definitions of relapse used in each individual study.
- In the table, an additional column with ‘measured relapse predictors’ should be added. In text, it would also be good to summarize the measured outcomes before answering the questions which predictors were significant.
- the discussion is mostly a more elaborate summary of the results and remains superficial. This discussion does not reflect an in depth discussion of the current study. Please put the outcomes of this study in a broader perspective.
Minor
- Abstract: a discussion/conclusion is missing. The results focus too much on the amount of literature found, and too little on the actual outcomes/answers on the research question.
- There is still a track-change in line 6 of the introduction (‘unacceptably’). Although I agree, this word is subjective, and I therefore suggest to remove. Throughout the paper, there are several typo’s.
Author Response
Responses to peer review comments:
Peer-reviewer 1:
This paper describes a systematic review focusing on predictors of Anorexia Nervosa relapse. This is an interesting topic, giving an important overview of the literature. However, I do belief some aspects could be further clarified to give a reader a better interpretation of the results:
- As the authors focus on relapse, a clear definition of relapse should be given, including clear in- and exclusion criteria related to this definition.
- We agree with this peer, and have added a definition in section 2.1. Importantly, we used the definition of relapse from each individual publication. All individual definitions have been added in Table 1, column 6 (definition of relapse), for clarity.
- Additionally, each included study used their own definition of relapse. These definitions needs to be described to help interpreting the results and drawing further conclusions.
- We agree and have added the definition of relapse for each individual publication in table 1, as mentioned above.
- In the description of study interventions, insights on duration, intensity, etc. should be given.
- We agree and have updated the table with the following information to the table 1:
- Duration of disease
- Severity of disease at start of treatment (if data available in the publication)
- The first time the research questions “what is the time to relapse” is mentioned is in section 3.2.4. this RQ should be introduced earlier already. Again, the answers given here depend on the definitions of relapse used in each individual study.
- We agree and have added this also in the Introduction, the Methods, results Table 1 and in the discussion.
- In the table, an additional column with ‘measured relapse predictors’ should be added. In text, it would also be good to summarize the measured outcomes before answering the questions which predictors were significant.
- We agree. It was however not possible to identify all predictors that potentially had been evaluated in all publications, and should this have been the case, this list would have been covering several pages. What we have done is to bold “identified predictors of relapse” from each included publication in Table 1, column 7.
- the discussion is mostly a more elaborate summary of the results and remains superficial. This discussion does not reflect an in depth discussion of the current study. Please put the outcomes of this study in a broader perspective.
- We have added several paragraphs in the discussion to address this.
Minor
- Abstract: a discussion/conclusion is missing. The results focus too much on the amount of literature found, and too little on the actual outcomes/answers on the research question.
- This has been corrected. We have added a conclusion after the discussion.
- There is still a track-change in line 6 of the introduction (‘unacceptably’). Although I agree, this word is subjective, and I therefore suggest to remove. Throughout the paper, there are several typo’s.
- This has been corrected.

Reviewer 2 Report
The analysis of the presented literature review is important to increasing the effectiveness of the treatment of anorexia nervosa. The methodology of selecting the research works used in this review has been presented quite accurately. However, with the publication base selected in this way, was it not possible to apply specific statistical techniques and qualitatively analyze the results observed in the presented works? Paying attention to the narrative aspect of the analysis exposes its subjective character. Wouldn't it be more appropriate to define a review as: systematic scoping review?
If the authors assumed that they only analyze the results of studies in which patients with AN achieved normal body weight, why explain some relationships with the fact that in selected studies not all patients had a BMI greater than 18.5 kg/m2?
It seems that for the analysis of the results of these selected studies, it would be interesting to compare them depending on the nature of the study, e.g. prospective vs. retrospective research, or depending on the interventions used to maintain remission of the disease, e.g. psychotherapy vs. psychotherapy and nutritional therapy.
There is no reference in the discussion to the mechanisms of the observed relationships regarding, for example, the causes of changes in body weight after its normalization or the influence of body composition and blood leptin concentration on changes in body weight in patients with AN.
The authors presented the strengths and weaknesses of their review but there was some lack of explanation of the strengths and weaknesses of the analyzed works.
In the purpose of review instead of "... BMI > 18.5 kg/m2." it should say "... BMI ³ 18.5 kg/m2."
Author Response
Response to Peer-reviewer 2
Comments and Suggestions for Authors
The analysis of the presented literature review is important to increasing the effectiveness of the treatment of anorexia nervosa. The methodology of selecting the research works used in this review has been presented quite accurately. However, with the publication base selected in this way, was it not possible to apply specific statistical techniques and qualitatively analyze the results observed in the presented works? Paying attention to the narrative aspect of the analysis exposes its subjective character. Wouldn't it be more appropriate to define a review as: systematic scoping review?
- We agree with this reviewer and it was our ambition to perform a meta-analysis of the outcomes e.g. rate of relapse and time-to relapse. However, data needed to perform a meta-analysis was missing, and also data to enable a regression or correlation analysis. We have added some basic statistics to the narrative synthesis of the results, which we believe have strengthened the results presentation and enabled a better conclusion of the results.
- We have in addition, since the manuscript largely presents the results in a narrative manner, added the term “scoping” to the title.
If the authors assumed that they only analyze the results of studies in which patients with AN achieved normal body weight, why explain some relationships with the fact that in selected studies not all patients had a BMI greater than 18.5 kg/m2?
- The inclusion criteria for the publications in this study was either BMI>5 at end of treatment, or a corresponding percentile if an adolescent population.
It seems that for the analysis of the results of these selected studies, it would be interesting to compare them depending on the nature of the study, e.g. prospective vs. retrospective research, or depending on the interventions used to maintain remission of the disease, e.g. psychotherapy vs. psychotherapy and nutritional therapy.
- We agree and have added a paragraph in the results section, providing a narrative comparison of the results from prospective vs retrospective. It was not the intention to compare interventions and during the process of reviewing publications, we excluded those that provided results from different interventions, as long as they did not enable us to extract information on weight gain alone.
- In addition, in the Discussion, we have added a paragraph dealing with the influence of study design.
There is no reference in the discussion to the mechanisms of the observed relationships regarding, for example, the causes of changes in body weight after its normalization or the influence of body composition and blood leptin concentration on changes in body weight in patients with AN.
- We agree and have added this in the discussion.
The authors presented the strengths and weaknesses of their review but there was some lack of explanation of the strengths and weaknesses of the analyzed works.
- We agree and have expanded the section in the Discussion on limitations.
In the purpose of review instead of "... BMI > 18.5 kg/m2." it should say "... BMI ³ 18.5 kg/m2."
- We have adjusted ”BMI > 18.5 kg/m2” to ”BMI >5 kg/m2”. We regard this as a most relevant improvement.

Round 2
Reviewer 1 Report
The authors did a good job in responding to my feedback